# Visual Data Diagnosis and Debiasing with Concept Graphs

**Rwiddhi Chakraborty**[1,2*] **Yinong (Oliver) Wang**[1] **Jialu Gao**[1] **Runkai Zheng**[1]
**Cheng Zhang**[1,3] **Fernando De la Torre**[1]

[1]Carnegie Mellon University  [2]UiT The Arctic University of Norway  [3]Texas A&M University

## Abstract

The widespread success of deep learning models today is owed to the curation of extensive datasets significant in size and complexity. However, such models frequently pick up inherent biases in the data during the training process, leading to unreliable predictions. Diagnosing and debiasing datasets is thus a necessity to ensure reliable model performance. In this paper, we present CONBIAS, a novel framework for diagnosing and mitigating **Con**cept co-occurrence **Bias**es in visual datasets. CONBIAS represents visual datasets as knowledge graphs of concepts, enabling meticulous analysis of spurious concept co-occurrences to uncover concept imbalances across the whole dataset. Moreover, we show that by employing a novel clique-based concept balancing strategy, we can mitigate these imbalances, leading to enhanced performance on downstream tasks. Extensive experiments show that data augmentation based on a balanced concept distribution augmented by CONBIAS improves generalization performance across multiple datasets compared to state-of-the-art methods.[2]

## 1 Introduction

Over the last decade we have witnessed an unparalleled growth in the capabilities of deep learning models across a wide range of tasks, such as image classification [17, 47, 7], object detection [41, 55], semantic segmentation [20, 26, 43], and so on. More recently, with the introduction of large multi-modal models, these capabilities have improved further [25, 15]. However, such models, while demonstrating impressive performance on a wide range of tasks, have been shown to be biased in their predictions [30, 13]. These biases come in various forms, based in texture [14], shape [39, 32], object co-occurrence [51, 52, 48], and so on. In addition to exploring model biases, dataset diagnosis, or evaluating biases directly within the dataset, is particularly crucial as large datasets available today are beyond the scope of human evaluation, owing to their size and complexity. For example, ImageNet [6], a widely used dataset in deep learning literature, is known to have thousands of erroneous labels and a lack of diversity in its class hierarchy [33, 58]. Other popular datasets such as MS-COCO [23] and CelebA [27], have problematic social biases with respect to gendered captions and prejudicial attributes of people from different races. As a result, frameworks that effectively diagnose and debias these datasets are sought.

While multiple works exist in the categorization and exploration of biases in visual data [9, 30], an end-to-end pipeline incorporating both diagnosis and debiasing has received relatively scant attention. ALIA [8] is the closest and most recent work exploring such a data-augmentation-based approach to debiasing, but it has two shortcomings - first, it does not diagnose the dataset which it aims to debias. Without such a diagnosis, it is challenging to identify the biases to be mitigated in the first place.

---

[*]Correspondence to: rwiddhi.chakraborty@uit.no
[2]Code: https://github.com/rwchakra/conbias

38th Conference on Neural Information Processing Systems (NeurIPS 2024).

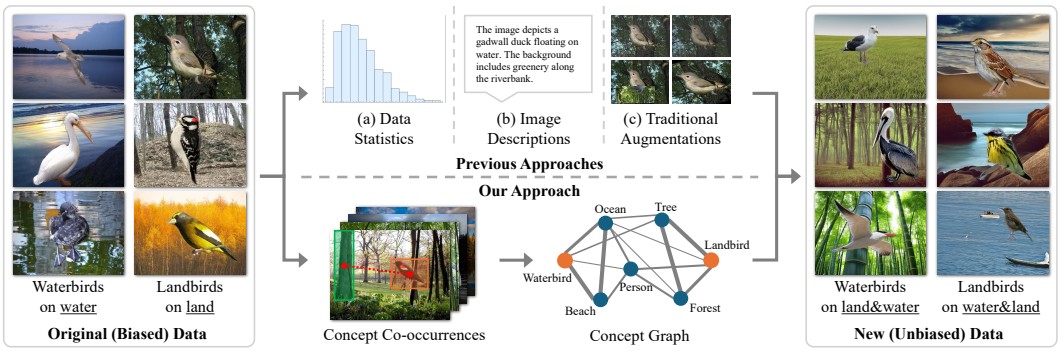

Figure 1: The conventional data diagnosis and augmentation pipeline begins with an original (biased) dataset. Existing methods address these biases via object frequency calibration [52], metadata analysis [8], or traditional augmentation techniques [60, 5]. In contrast, our framework models visual data as a knowledge graph of concepts, with orange nodes representing classes and blue nodes representing concepts, facilitating a systematic diagnosis of class-concept imbalances for debiasing object co-occurrences in vision datasets.

Second, the method relies on a large language model (ChatGPT-4 [4]) to generate diverse, unbiased, in-domain descriptions. This approach is potentially confounding since there is no reliable way to ensure that the biases of the large language model itself do not affect the quality of such domain descriptions. In this work, we address both these shortcomings.

We present CONBIAS, our framework for diagnosis and debiasing of visual data. Our key contribution is in representing a visual dataset as a knowledge graph of concepts. Analyzing this graph for imbalanced class-concept combinations leads to a principled diagnosis of biases present in the dataset. Once identified, we generate images to address under-represented class-concept combinations, promoting a more uniform concept distribution across classes. By using a concept graph, we circumvent the reliance on a large language model to generate debiased data. Figure 1 illustrates the core idea of our approach in contrast with existing methods. We target object co-occurrence bias, a human-interpretable issue known to confound downstream tasks [34, 10]. Object co-occurrence bias refers to any spurious correlation between a label and an object causally unrelated to the label. Representing the dataset as a knowledge graph of object co-occurrences provides a structured and controllable method to diagnose and mitigate these spurious correlations.

Our framework proceeds in three steps: (1) *Concept Graph Construction:* We construct a knowledge graph of concepts from the dataset. These concepts are assumed to come from dataset ground truth such as captions or segmentation masks. (2) *Concept Diagnosis:* This stage then analyzes the knowledge graph for concept imbalances, revealing potential biases in the original dataset. (3) *Concept Debiasing:* We sample imbalanced concept combinations from the knowledge graph using graph cliques, each representing a class-concept combination identified as imbalanced. Finally, we generate images containing under-represented concept combinations to supplement the dataset. The image generation protocol is generic and uses an off-the-shelf inpainting process with a text-to-image generative model. This principled approach ensures that the concept distribution in our augmented data is uniform and less biased. Our experiments validate this approach, showing that data augmentation based on a balanced concept distribution improves generalization performance across multiple datasets compared to existing baselines. In summary, our contributions include:

- We propose a new concept graph-based framework to diagnose biases in visual datasets, which represents a principled approach to diagnosing datasets for biases, and to mitigating them.
- Based on our graph construction and diagnosis, we propose a novel clique-based concept balancing strategy to address detected biases.
- We demonstrate that balanced concept generation in data augmentation enhances classifier generalization across multiple datasets, over baselines.

## 2 Related Work

**Bias discovery in deep learning models.** The identification of biases in trained deep learning models has a rich history, with early works exploring the texture and shape-bias tradeoff in ImageNet-

pretrained ResNets [14, 21, 32, 39]. More recently, the field of worst group robustness has emerged, aiming to generalize classifier performance across multiple groups in the data that correspond to known spurious correlations [49, 45, 24, 42]. Debiasing and concept discovery in the feature space of the learned classifier is also common [1, 54, 59]. Testing model performance sensitivity to the presence of particular attributes has also been explored [53, 36]. With the recent rise in popularity of large language models, efforts have been made to identify learned biases using off-the-shelf captioning models [56], adaptive testing [11], and language guidance [19, 37]. Traditional data augmentation approaches such as CutMix [60], and RandAug [5], are used as baselines as well. Our work intervenes on the dataset directly, instead of operating in the model feature space or testing model sensitivity. This allows for a more intuitive and principled approach to bias discovery.

**Data diagnosis.** Our work is placed in the context of data diagnosis, i.e. identifying biases directly from the data without using the model as a proxy. One of the early influential works expounding the importance of datasets in deep learning research was a systematic review of the popular datasets in computer vision [51]. A modern appraisal categorizing more diverse types of biases in visual datasets exists in [9]. Additionally, works investigating possible issues with dataset labels have also received interest [33, 58]. Data diagnosis tools such as REVISE [52] compute object statistics (including co-occurrence) to generate high-level insights of the data. However, REVISE is not an end-to-end framework that at once diagnoses and debiases data. It is rather an exploratory tool for an overview of common concepts in the dataset. A more recent method, ALIA, uses a language model to populate diverse descriptions of the given dataset, consequently generating images from such descriptions. A more critical look on dataset bias lies in the field of fairness, particularly with regards to societal bias [12, 16]. Finally, benchmark datasets for data diagnosis have also been proposed [29, 28].

**Object co-occurrence bias in visual recognition.** Objects are biased in the company they keep. This adage is well known in the computer vision literature, as outlined in [34, 10]. Modern efforts to mitigate object co-occurrence bias involve feature decorrelation [48], object aware contrastive representations [31], causal interventions [38], and fusing object and contextual information via attention [2]. The common theme in tackling contextual and co-occurrence bias lies entirely in using better models (feature representations) rather than intervening in the dataset directly. We place our debiasing method along the data augmentation direction, allowing for better controllability and interpretability of the debiasing stage, rather than relying on semantic features learned by a classifier, which may be difficult for humans to interpret.

## 3 Approach

Figure 2 illustrates the overall pipeline of our method. In this section, we begin with the problem statement in Section 3.1, and move to the three major stages in our method definition. Section 3.2 describes the procedure of concept graph construction. Section 3.3 illustrates the details of concept diagnosis. Finally, Section 3.4 presents our method for concept debiasing.

### 3.1 Problem Statement

We are given a dataset $D = \{(x_i, y_i)\}_{i=1}^N$, a set of images and their corresponding labels. We also assume access to a concept set $C = \{c_1, c_2, \ldots, c_k\}$ that describes unique objects present in the data. An example concept set looks like the following: {alley, crosswalk, downtown, ..., gas station}, i.e. a list of unique objects present in each image in addition to the class label. Finally, we are given a classifier $f_\theta(X)$ parameterized by network parameters $\theta$. The central hypothesis of this work is that the class labels exhibit co-occurring bias with the concept set $C$, affecting downstream task performance. In this light, we wish to generate an augmented dataset $D_{\text{aug}}$ that is debiased with respect to the concepts and their corresponding class labels. Thus, given the new dataset $D' = D \cup D_{\text{aug}}$, we wish to retrain $f_\theta(X)$ in the standard classification setup:

$$\hat{f}^* = \arg\min_f \mathbb{E}_{(x,y) \in D'}[\mathcal{L}(y, f_\theta(x))], \tag{1}$$

where $\mathcal{L}(y, f_\theta(x))$ is the cross entropy loss between the class label and classifier prediction. Our framework consists of three stages: *Concept Graph Construction*, *Concept Diagnosis*, and *Concept Debiasing*. Next, we provide details on each step.

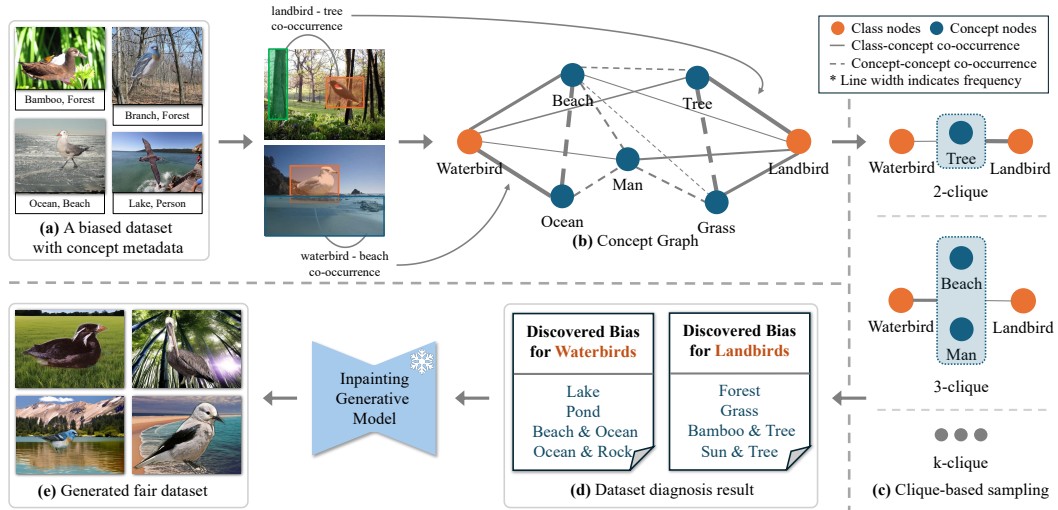

Figure 2: **Overview of our framework** CONBIAS. (a) Given a dataset and its concept metadata which contains the objects present in each image, (b) we build the concept graph using object co-occurrences. The line thickness indicates the co-occurrence frequencies of particular concepts with their respective classes. (c) Next, the clique-based sampling strategy generates under-represented class-concept combinations, which yield (d) the dataset diagnosis result. (e) Finally, with biases discovered, we generate images of classes containing under-represented concept combinations in the dataset with a standard text-to-image generative model.

## 3.2 Concept Graph Construction

We construct a concept graph $G = (V, E, W)$ from the data, where $|V|$ is the node set of the graph, $|E|$ is the edge set, and $|W|$ is the set of weights for each edge in the graph. We first construct the node set $V$ as a union of the label set $Y$ and concept set $C$:

$$V = Y \cup C.$$

Next, we construct the edge set $E$:

$$E = \{(i, j) \mid \exists \, \text{image } D_k \text{ such that both } i \text{ and } j \text{ appear in } D_k\}.$$

Finally, we construct the weight set $W$ by computing the weights $w_{ij}$ for each edge $(i, j)$ in $G$:

$$w_{ij} = \sum_{n=1}^{N} \mathbb{I}(i \in D_n \text{ and } j \in D_n),$$

where $\mathbb{I}$ is the indicator function that returns 1 if both $i$ and $j$ are present in the $n$-th image in $D$, and 0 otherwise, and $N$ is the total number of images in the dataset.

The concept graph $G$ encapsulates co-occurrence counts between nodes, thus providing an alternative representation of the (originally visual) data. As we show in the next section, this representation helps uncover novel imbalances (bias) contained in the dataset.

## 3.3 Concept Graph Diagnosis

In the previous section, we define how to build the concept graph. Here, we present how to leverage the concept graph for discovering co-occurrence biases. We present a principled approach to discovering concept-combinations across classes that co-occur in an imbalanced fashion.

**Definition (Class Clique Sets)**   For each class $Y_i \in Y$, we construct a set of $k$-cliques using the concept graph $G$. The set of all possible $k$-cliques for class $Y_i$ is denoted as $\mathcal{K}_i^k$:

$$\mathcal{K}_i^k = \{\{c_{j_1}, c_{j_2}, \ldots, c_{j_k}\} \mid c_{j_1}, c_{j_2}, \ldots, c_{j_k} \in C \text{ and } j_1 < j_2 < \ldots < j_k\},$$

where $j_1, j_2, \ldots$ are the indices of concepts in $C$. Then, $\mathcal{K}_i$ for class $Y_i$ can be successfully constructed for $k = 1, 2, \ldots, K$, where $K$ is the size of the largest clique in $G$ containing $Y_i$. We construct class clique sets for every class in the dataset. An illustration of concept cliques in the Waterbirds dataset that help in bias diagnosis is provided in Figure 3.

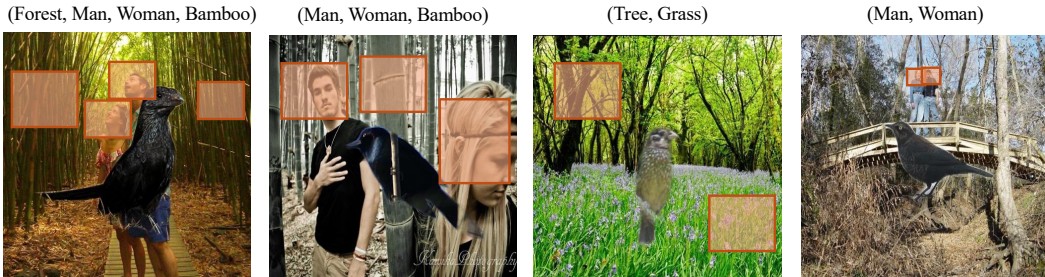

| (Forest, Man, Woman, Bamboo) | (Man, Woman, Bamboo) | (Tree, Grass) | (Man, Woman) |

Figure 3: Examples of concept clique sets for `Landbird` class in Waterbirds dataset uncovered by our diagnosis. Concepts such as `Tree`, `Forest`, `Man`, `Woman`, `Bamboo` are overwhelmingly associated with this class, indicating strong co-occurrence bias. All these concepts are causally unrelated to the bird type.

**Definition (Common Class Clique Sets)**   Given $\mathcal{K}_i$ for each class, we then compute the cliques common to all classes. These are the cliques of interest, whose imbalances we want to investigate:

$$\mathcal{K} = \bigcap_i \mathcal{K}_i,$$

where $\mathcal{K}$ encapsulates all common cliques enumerated across the dataset for all classes. Refer to Figure 2 for a broad illustration of the $k$-clique set construction from the concept graph $G$.

**Definition (Imbalanced Common Cliques)**   Given the set of common cliques across all classes $\mathcal{K}$, we compute the imbalanced class-concept combinations, i.e. the imbalanced clique set $I$:

$$I_{[\mathcal{K}]_{m=1}^M} = \{(|F_{\mathcal{K}_{y_i}^m} - F_{\mathcal{K}_{y_j}^m}|, \arg\min(F_{\mathcal{K}_{y_i}^m}, F_{\mathcal{K}_{y_j}^m}))\}, \forall i, j,$$

where $F_{\mathcal{K}_{y_i}^m}$ and $F_{\mathcal{K}_{y_j}^m}$ indicates the co-occurrence frequency of concepts in clique $m$ with respect to class $y_i$ and $y_j$ respectively, and the $\arg\min$ operator identifies the underrepresented class for the particular concept clique. Thus, each element in $I$ is a number representing the imbalance of each common clique across all classes. For the special case where the size of clique $m$ is 1, this equates to simply looking up the value $w_{ij}$ in $G$. For the case where the size of $m > 2$, it is straightforward to compute the co-occurrence of class $y_i$ with respect to concepts in $m$:

$$w_{ij\ldots k} = \sum_{n=1}^N \mathbb{I}(i \in D_n \text{ and } j \in D_n \ldots \text{ and } k \in D_n),$$

for each image $D_n$ in the data. The set $I$ holds rich information about the data. In addition to holding the imbalanced counts of concept combinations across all classes, the set $I$ also holds which is the *underrepresented* class with respect to a particular concept clique.

Intuitively, concept combinations that are common across all the classes, but do *not* co-occur uniformly across the classes are likely biased concept combinations. We provide an example from the *Waterbirds* dataset in Figure 4. The training set in *Waterbirds* is intentionally biased to the background: 95% of landbirds appear with land backgrounds, and 95% of waterbirds appear with water backgrounds. First, we find common cliques of varying sizes across the classes (`Landbird`, `Waterbird`). One example of a common clique of size 3 is (`Landbird`, `Beach`, `Ocean`) and

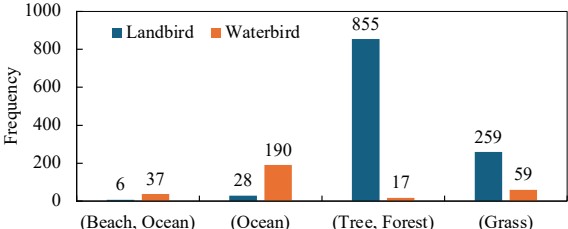

Figure 4: **Examples of concept imbalances in the Waterbirds dataset.** We show the frequencies of concepts cliques as discovered in the dataset. We see imbalances across not only single concepts (e.g., `Ocean`, `Grass`) but also concept combinations (e.g., (`Beach`, `Ocean`), (`Tree`, `Forest`)). These are the biases we aim to mitigate for the downstream task.

(`Waterbird`, `Beach`, `Ocean`). We compute the co-occurrence of (`Landbird`, `Beach`, `Ocean`) and (`Waterbird`, `Beach`, `Ocean`) from the extracted metadata, and the imbalance is clear. Since waterbirds are far more prone to appear on water, there are significantly more images of waterbirds

containing concepts `Beach` and `Ocean` than landbirds, which are more prone to be in land-based environments. If we look at the co-occurence of `Landbird` with a land-based concept such as `Grass`, we see the opposite imbalance. There are significantly more images of landbirds containing trees over waterbirds. Similarly, for the water-based concept of `Ocean`, we see a strong imbalance towards the `Waterbird` class. In our debiasing stage, we should therefore generate more images of waterbirds with tree-based backgrounds, and landbirds with beach/water-based backgrounds. Using the clique-based approach, we have successfully uncovered the known background bias in the Waterbirds dataset. This approach is generalizable to multiple classes. All we need are common cliques, and the computation of concept co-occurrences across the dataset. In this way, our concept graph approach uncovers interesting concept combinations across the *whole* dataset that appear in an imbalanced and spurious fashion. More examples of such imbalances are provided in the supplementary material.

### 3.4   Concept Graph Debiasing

We have, to this point, constructed a knowledge graph of the visual data, and diagnosed it for concept-based co-occurrence biases. Once the imbalanced clique set $I$ is identified in $G$, we debias the data by generating images containing under-represented concepts across classes.

Recall that $I = \{f_i, Y_i\}$ inherently holds the underrepresented class $Y_i$ and the frequency $f_i$ by which the original dataset needs to be adjusted with new images of class $Y_i$ with respect to concept clique $i$. Following the example in the previous section, we notice that the concepts (`Beach, Ocean`) are significantly over-represented in the `Waterbird` class than `Landbird`. Similarly, the concept `Tree` is significantly over-represented with the `Landbird` class than the `Waterbird` class. As a result, we sample $f_i$ instances of these under-represented cliques with respect to their classes, and prompt a text-to-image generative model for more images of the `Waterbird` class with the concept `Tree`. Similarly, we would prompt the model to generate images of Landbird with the concept `Beach, Ocean`. We generate images for all class based imbalances following this upsampling protocol. Typical prompts for our image-inpainting model would look like: `An image of a ocean and a beach`, `An image of a tree`, `An image of a forest`, etc. We use an inpainting-based method to make sure that the original object is not modified in the image, and that the new concepts are only injected into the non-object space in the image. See the supplementary material for the generated images and the prompts.

Using this upsampling protocol, we generate a set of images that leads to our augmented, debiased dataset $D_{\text{aug}}$. The original training data $D$ can now be augmented using this data, and the classifier $f_\theta(X)$ can be retrained on the dataset $D \cup D_{\text{aug}}$. In the next section, we conduct experiments on three datasets to demonstrate our method's significant improvements of baselines.

**Note on concept set annotations.** We assume the availability of reliable ground truth concept sets. Such annotations already exist for the datasets we investigate - Waterbirds, UrbanCars, and COCO-GB. We agree that unreliable ground truth concept sets would hinder generalization abilities, but this assumption is not dissimilar to the assumption of reliable ground truth labels in classification tasks. Moreover, the reliance on ground truth concept sets, sometimes referred to as concept banks, have also been considered in prior work [57]. Ground truth concept sets serve as auxiliary knowledge bases and provide human level interpretability to the task at hand.

**Note on computational complexity.** In general, given $K$ classes and $C$ concepts, the graph clique enumeration is expected to grow in $O(exp(K + C))$. However, in practice, we find that constraining clique sizes $\leq 4$ leads to interpretable bias combinations, with no significant effect of the exponential runtime.

## 4   Experiments

We validate our method on vision datatset diagnosis and debiasing across various scenarios. We begin by introducing the experimental setup including the datasets, baselines, tasks, and implementation details in Section 4.1. Section 4.2 presents the main results of our proposed framework, CONBIAS, compared with state-of-the-art methods. Finally, Section 4.3 details ablation studies and analyses.

Table 1: **State-of-the-art comparison on different datasets.** Results are averaged over three training runs. **CB**: class balanced split. **OOD**: out-of-distribution split. Binary class classification accuracy is used as the metric. Our method outperforms previous approaches across multiple datasets.

| Method | Waterbirds [45] | | UrbanCars [22] | | COCO-GB [50] | |
|---|---|---|---|---|---|---|
| | CB ↑ | OOD ↑ | CB ↑ | OOD ↑ | CB↑ | OOD ↑ |
| Baseline [17] | 67.1 | 44.9 | 73.5 | 40.5 | 58.5 | 51.9 |
| + RandAug [5] | 73.7 | 60.2 | 76.3 | 46.1 | 55.8 | 50.2 |
| + CutMix [60] | 67.9 | 45.6 | 74.4 | 39.3 | 57.4 | 51.2 |
| + ALIA [8] | 69.6 | 48.2 | 74.0 | 42.5 | 58.7 | **52.4** |
| + CONBIAS (Ours) | **77.9** | **69.3** | **78.3** | **52.9** | **58.8** | 51.4 |

## 4.1 Setup

**Datasets.** We use three datasets in our work: Waterbirds [45], UrbanCars [22], and COCO-GB [50], that are commonly used in the bias mitigation domain. We tackle background bias in the Waterbirds dataset, background and co-occuring object bias in the UrbanCars dataset, and finally gender bias in COCO-GB. All the tasks are binary classification tasks. More details on the training splits and class labels are provided in the supplementary material. For Waterbirds, there are 66 nodes and 865 edges in the concept graph. For UrbanCars, the graph contains 19 nodes and 106 edges. For COCO-GB, there are 81 nodes and 2326 edges in the graph.

**Baseline methods.** Our baselines are include a vanilla Resnet-50 model pre-trained on ImageNet, two typical data augmentation based debiasing methods: (1) CutMix, a technique where we cut and paste patches between different training images to generate diverse discriminative features, and (2) RandAug, which creates random transformations on the training data during the learning phase. Finally, we compare against the recently proposed and state-of-the-art ALIA[8], which uses a large language model to generate diverse, in-distribution prompts for a text-to-image generative model.

**Evaluation protocols.** We compute the mean test accuracy over the class-balanced test data and the out-of-distribution (OOD) test data, similar to [8]. The class-balanced data contains an even distribution of classes and their respective spurious correlations, while the OOD data contains counterfactual concepts. For example, in Waterbirds dataset, for the class-balanced test data 50% images of Landbirds have Land backgrounds, while 50% images of Waterbirds have Water backgrounds. The OOD test set contains Landbirds on Water, and Waterbirds on Land. More details on the test sets are presented in the supplementary material.

**Implementation details.** We use existing implementations to train our models. Our Base model is a Resnet-50 pretrained on ImageNet [17]. We generate the same number of images per data-augmentation protocol to ensure a fair comparison. For comparison with ALIA on Waterbirds, we directly use their generated dataset available here. For the other datasets, we used the existing ALIA implementation to generate the augmented data. Following previous work, we use validation loss based checkpointing to choose the best model, the Adam optimizer with a learning rate of $10^{-3}$, a weight decay of $10^{-5}$, and a cosine learning schedule over $100$ epochs. To generate images, we use Stable Diffusion [44] with a CLIP [40]-based filtering mechanism to ensure reliable image generation. Finally, we inpaint the object onto the generated image using ground truth masks (available for all datasets). All code was written in PyTorch [35].

## 4.2 Main Results

In Table 1 we present the main results, averaged over three training runs. First, we note that for Waterbirds and UrbanCars, we observe significant improvements in both the Class-Balanced and OOD test sets over the typical augmentation methods such as CutMix and RandAug. Second, we note the significant improvement in performance over the most recent state-of-the-art augmentation method, ALIA. Third, for COCO-GB, while we notice slightly smaller difference in the CB and OOD accuracies between our method and the baselines, our hypothesis is that this happens because of limited number of samples used for augmentation. ALIA uses a confidence based filtering mechanism to remove generated samples. This leads to a small final number of 260 samples to be added for the

Table 2: **Benefit of the graph structure in** CON-BIAS. Leveraging the graph structure is beneficial as opposed to simply computing single concept-class frequency counts on UrbanCars.

| Model | CB ↑ | OOD ↑ |
|---|---|---|
| Base | 73.4 | 40.4 |
| Base + ALIA | 74.0 | 42.5 |
| Base (BG) | 78.5 | 51.9 |
| Base (CoObj) | 77.0 | 47.3 |
| Base (Both) | 78.1 | 51.3 |
| Base (CONBIAS) | **79.4** | **53.2** |

Table 3: **Performance for the IP2P variant** of CONBIAS with respect to base, ALIA, and our original model on Waterbirds. Our method significantly improves over ALIA even when using IP2P, although the best results are still achieved when using the stable diffusion based inpainting protocol.

| Models | CB ↑ | OOD ↑ |
|---|---|---|
| Base | 67.1 | 44.9 |
| Base + ALIA | 69.6 | 48.2 |
| Base + CONBIAS (IP2P) | 72.9 | 60.5 |
| Base + CONBIAS | **77.9** | **69.3** |

Table 4: **Robustness of our method to evaluation metrics** In addition to CB and OOD performance, we also report metrics evaluating multiple shortcut mitigation. Results on UrbanCars (Average of three training runs).

| Model | BG-GAP ↑ | CoObj-GAP ↑ | BG+CoObj GAP ↑ |
|---|---|---|---|
| Base | -11.2 | -21.5 | -54.8 |
| Base (BG) | **-5.0** | -19.4 | -38.0 |
| Base (CoObj) | -6.3 | -19.2 | -47.3 |
| Base (Both) | -5.6 | -23.2 | -47.6 |
| Base (CONBIAS) | -6.0 | **-18.4** | **-41.4** |

retraining part. In the ablation section, we show this hypothesis to be true, and further demonstrate that on adding more images for the retraining step, we progressively increase the performance gap between our method and the baselines. These three observations taken together validate the usefulness of our approach. The next section provides additional insights on the usefulness of our method and the effect of ablating its components.

### 4.3 Ablations and Analyses

We further analyze our method along five axes: (1) The usefulness of the graph structure, (2) Robustness of our method to other evaluation metrics, (3) The impact on CB and OOD performance by increasing the number of added samples for the retraining step, (4) The usefulness of discovered concepts by our method on the trained classifier, and (5) The impact of the generative component in our work compared to ALIA, since the latter uses InstructPix2Pix [3] while we use a Stable Diffusion based inpainting protocol.

**Effect of the graph structure.** Recalling the definition of Class Clique Sets, in principle one could only use cliques sizes of 1, i.e., the direct neighbors of each class node. This would be equivalent to computing the frequency of co-occurrence over a single hop neighborhood of the class node in the graph. In this ablation we show that one should use larger cliques, i.e. leverage the graph structure, instead of a simple direct neighborhood based frequency calculation. We trained three separate models on three different types of $D_{aug}$: Ours (BG), trained on images containing only background shortcuts, Ours (CoObj), images containing only the co-occurrence shortcuts, and Ours (Both), images containing both shortcuts, but *not simultaneously*.

Table 2 shows the results. First, our approach of leveraging the graph structure provides improvement over simply using the frequency of a 1-hop neighborhood. Second, we note that *all* the methods outperform the baseline and ALIA, which shows that incorporating frequency based co-occurrences is in a broader sense much more useful than relying on diverse prompts generated by ChatGPT-4, which is the approach taken by ALIA.

**Robustness to evaluation metrics.** The CB and OOD test accuracies test for generalization capabilities, but more direct evaluators of shortcut learning exist in the literature. In [22], for instance, the authors propose (i) The *ID Accuracy* - which is the accuracy when the test set contains common background and co-occurring objects, (ii) The *BG-GAP* - which is the drop in *ID accuracy* when the test set contains common co-occurring objects, but uncommon background objects, (iii) The

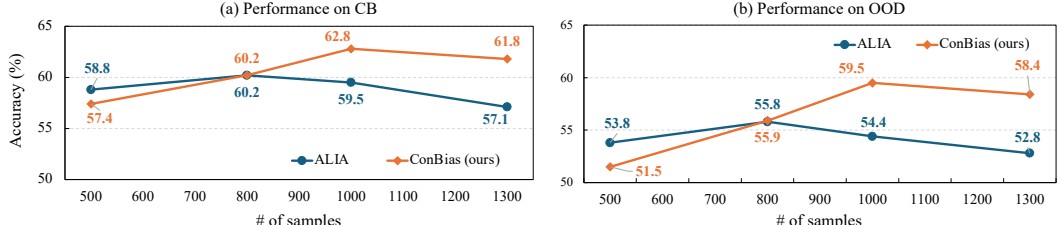

Figure 5: **Performance on COCO-GB.** We show the accuracies on (a) Class-Balanced (CB) and (b) Out-of-Distribution (OOD) splits. We observe that increasing number of images in $D_{\text{aug}}$ improves performance up to a certain point (1000 images).

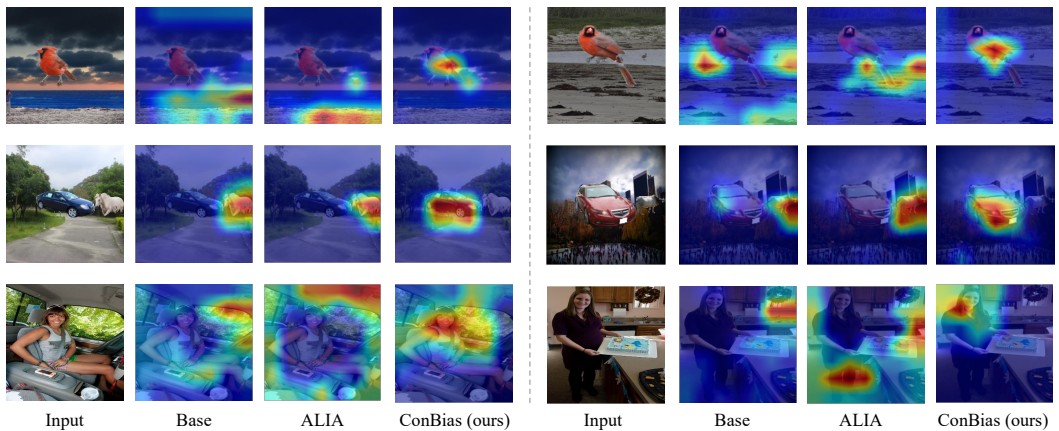

Input     Base     ALIA     ConBias (ours)       Input     Base     ALIA     ConBias (ours)

Figure 6: **Visualization of the heatmaps for different methods.** Top row: Waterbirds. Middle row: UrbanCars. Bottom row: COCO-GB. Our method enforces the base model to focus on only the relevant features in the data, and removing reliance on shortcut features, i.e. the background for Waterbirds, the background and co-occurring object for UrbanCars, and the gender for COCO-GB.

*CoObj-GAP*, which is the drop in *ID accuracy* when the test set contains uncommon co-occurring objects, but common background objects, and finally (iv) The *BG + CoObj GAP*, which is the case when both background and co-occurring objects are uncommon in the test set. A multiple shortcut mitigation method should *minimize* the *BG + CoOBj GAP* metric, and also make sure it does not exacerbate any shortcut that the base model relies on. In Table 4, we present results of Base, Base (BG), Base (CoObj), Base(Both), and Base (CONBIAS) on these metrics for UrbanCars. We are able to post the lowest drops among all baselines on the *CoObj-GAP* and *BG + CoOBj GAP* metrics, suggesting mitigation of multiple shortcut reliance. This places our method in a more realistic context, as it is infeasible to assume that real world data will only have a single type of bias in them.

**Scaling the number of images in $D_{\text{aug}}$.** In Table 1, we commented on the fact that our method provides marginal improvement over the baselines in the COCO-GB dataset. Our hypothesis was that this was due to the low number of images in the augmented dataset. In Figure 5, we demonstrate the impact of adding more images to $D_{\text{aug}}$ for retraining. Clearly, our method benefits from this protocol, leading to significant differences over ALIA as we keep increasing the number of images. Note that, infinite enrichment is not recommended and has been found to be detrimental to classifier performance, as progressive addition of synthetic images will likely lead to addition of out-of-distribution examples in the training data. This explains why, after an inflexion point, the accuracy suffers from adding more images. Similar observations have been made in [8] and [18].

**Discovered concept imbalances and feature attributions.** To verify that the model indeed debiases the imbalanced concepts that our method discovers, we present GradCAM [46] attributions of the model predictions after retraining. In Figure 6, we show results on all datasets. While other methods frequently focus on the spurious feature , CONBIAS helps the model focus only on the relevant, object level features of the data.

**The impact of the generative model.** ALIA uses an InstructPix2Pix (IP2P) based generation procedure, while we use stable diffusion with a mask-inpainting procedure to make sure the objects remain consistent in the image. To ablate the effect of the generation, we present results of our method with IP2P as the generative model instead, on Waterbirds dataset, in Table 3. First, we note that even with IP2P as the generative component, we are able to outperform ALIA, which suggests that it is actually the superior quality of our concept discovery method that leads to the improved results. Second, our inpainting based method outperforms our IP2P based method, which we argue is due to the objects being preserved in the generated image, as opposed to traditional image editing methods, where the object may transform arbitrarily, hurting the quality of augmented data.

## 5 Conclusion, Limitations, and Future Works

While CONBIAS is the first end-to-end pipeline to both diagnose and debias visual datasets, there are some limitations: First, that the enumeration of cliques grows exponentially with the size of the graph. For larger real world graphs, there could be more efficient strategies to find the concept combinations. Second, in this work we focus on biases emanating out of object co-occurrences. A variety of other biases exist in vision datasets, and future work would look to address the same. We add an extended section on broader impact of our work in the supplementary material. In summary, datasets in the real world are biased, and the exponential increase in dataset sizes over the past decade amplifies the challenge of investigating model and dataset biases. While both dataset and model diagnosis are exciting areas of research, an end-to-end diagnosis and debiasing pipeline such as CONBIAS offers a principled approach to diagnosing and debiasing visual datasets, in turn improving downstream classification performance. Our state-of-the art results open up numerous interesting possibilities for future work - incorporating more novel graph structures, and diagnosis under the regime where concept sets may be wholly or partially unavailable, remain interesting directions to pursue.

## Acknowledgements

This research is partially supported by a grant from Apple Inc. We thank Nicholas Apostoloff, Oncel Tuzel, and Jerremy Holland for their valuable feedback on the draft of this paper. Any views, opinions, findings, and conclusions or recommendations expressed in this material are those of the authors and should not be interpreted as reflecting the views, policies or position, either expressed or implied, of Apple Inc. The authors would also like to thank the Norwegian Research Council for funding a doctoral research visit to the Human Sensing Lab, Carnegie Mellon University.

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

## Supplementary Materials

In this supplementary materials, we provide details and additional results omitted in the main text.

## 6   Broader Impact

Fairness in AI is rapidly gaining priority in current research as models and datasets grow exponentially larger, thus making it more and more complicated to diagnose them for biases. It is imperative to focus on understanding and mitigating biases learned by models, and inherent biases in the data, to ensure reliable and transparent predictions in the real world. The advent of generative models in particular, including large language models, and image generative models, invites new questions into how to reliably regulate such technologies. These models are trained on datasets in the order of hundreds of billions of data points. How do we ensure that problematic aspects of the data do not pass onto the models learning from them? How do we ensure that models do not generate synthetic data that is potentially harmful, misleading, and misinformative in nature? How do we evaluate the quality of generated data by such models? These are the pressing questions that our research direction is interested in.

## 7   Dataset Details

We use three datasets in our work - Waterbirds [45], UrbanCars [22], and COCO-GB [50].

For Waterbirds, the class labels are *Landbird, Waterbird*. The Waterbirds dataset has the background bias, i.e. 95% images of landbirds have land-based backgrounds, and 95% images of waterbirds have water-based backgrounds. For the concept set annotations, we use the captions extracted by authors of [8] captions available here.

For UrbanCars, the class labels are *Urban, Country*, defining the type of car. There are multiple biases in UrbanCars - (1) Background Bias, i.e. Urban cars appear with 95% correlation with urban backgrounds, and Country cars appear with 95% correlation with country backgrounds. (2) Co-Occurring object, i.e. Urban cars appear with 95% correlation with urban objects, and Country cars appear with 95% correlation with country objects.

For COCO-GB, the class labels are *Man, Woman*. The bias for the dataset are the set of objects in the MS-COCO dataset [23]. The authors of [50] find a strong bias of most objects in the data with respect to the "Man" class, and design a secret, gender-balanced test set to evaluate gender bias in classifiers.

### 7.1   Waterbirds

In Fig 7 we present examples from the Waterbirds training data. The classes are heavily biased to the backgrounds, i.e. Landbirds on Land, Waterbirds on Water.

### 7.2   UrbanCars

In Fig 8 we present examples from the UrbanCars training data. The classes are heavily biased to multiple shortcuts - Background and Co-Occurring objects.

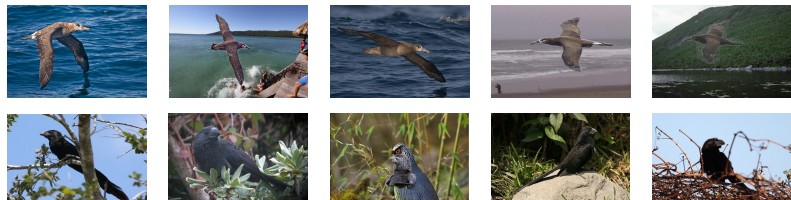

Figure 7: Examples of training data in Waterbirds dataset. Waterbirds (Top) are 95% biased towards water backgrounds, while Landbirds (Bottom) are 95% biased towards land backgrounds.

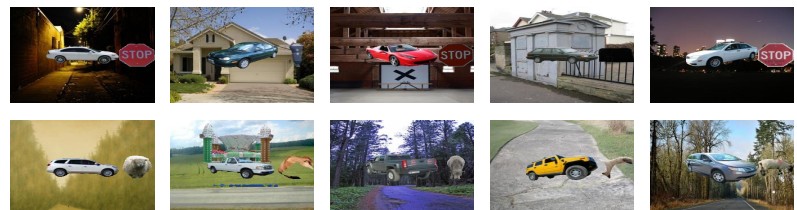

Figure 8: Examples of training data in UrbanCars dataset. Urban cars (Top) are 95% biased towards urban backgrounds and urban co-occurring objects. Country cars (Bottom) are 95% biased towards country backgrounds and country co-occurring objects.

### 7.3 COCO-GB

In Fig 9 we present examples from the COCO-GB training data. The "Man" class is known to be heavily biased in MS-COCO to everyday objects.

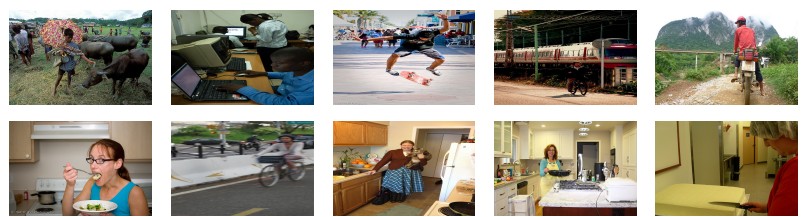

Figure 9: Examples of training data in MS-COCO dataset. Images of men are heavily biased towards common, everyday objects, as opposed to women. Authors of [50] find over a 90% in all object correlations towards men.

### 7.4 Splits

In Table 5 we present the train, validation, and test splits for our three datasets.

| Dataset | Train | Test | Validation |
|---------|-------|------|------------|
| Waterbirds | 4795 | 1199 | 5794 |
| UrbanCars | 8000 | 1000 | 1000 |
| COCO-GB | 32582 | 1331 | 1000 |

Table 5: Dataset sizes for Train, Test, and Validation sets

## 8 Concept Sets

In Table 6 we present the full concept sets for each dataset. The Waterbirds dataset has 64 unique concepts, the UrbanCars dataset has 17 unique concepts, and COCO-GB has 81 unique concepts, all

from the MS-COCO dataset. Note that both MS-COCO and UrbanCars have ground truth concepts, while for Waterbirds, we use the extracted captions here.

| Dataset | Concepts |
|---------|----------|
| Waterbirds | duck, pond, tree, grass, post, ocean, bridge, surfer, surfboard, beach, people, forest, beak, sailboat, bamboo, sunlight, boy, foot, boat, mountain, seagull, field, rock, crab, wall, woman, cell phone, man, wing, deer, leaf, backpack, hillside, statue, display, wave, lake, pen, palm tree, shirt, sign, bamboo forest, grass plant, tree branch, bushes, horse, sidewalk, parrot, sun, cup, town, snowy forest, red eye, twig, wooden fence, path, penguin, fishing rod, pelican, kayak, wine glass, lighthouse, mountain landscape, wooden path |
| UrbanCars | alley, crosswalk, downtown, gas station, garage-outdoor, driveway, forest road, field road, desert road, fireplug, stop sign, street sign, parking meter, traffic light, cow, horse, sheep |
| COCO-GB | stop sign, tie, knife, car, bicycle, fire hydrant, cow, motorcycle, umbrella, sports ball, cat, surfboard, elephant, skateboard, skis, backpack, couch, bed, wine glass, carrot, cup, airplane, handbag, cake, cell phone, woman, refrigerator, potted plant, sandwich, vase, chair, bus, frisbee, parking meter, bench, horse, truck, snowboard, train, clock, keyboard, scissors, man, bottle, kite, traffic light, book, dining table, sheep, fork, spoon, tennis racket, dog, bowl, suitcase, boat, donut, baseball bat, orange, toothbrush, banana, oven, laptop, toilet, sink, pizza, mouse, baseball glove, tv, teddy bear, hot dog, broccoli, remote, bird, microwave, apple, zebra, bear, toaster, giraffe, hair drier |

Table 6: Concepts for Waterbirds, UrbanCars, and COCO-GB datasets

## 9    Dataset Imbalances

In this section we shed more insight into what sort of concept imbalances ConBias discovers. These object level insights are also, to the best of our knowledge, the first of its kind, shedding more light on the secret co-ocurrence biases hidden in data.

### 9.1    Waterbirds

In addition to the main paper, we list some other category imbalances in Waterbirds in Figure 10. Some of these extreme imbalances appear in diverse 2-clique/3-clique combinations. For example, we see that concepts like forest, man, woman are significantly biased towards the Landbird class, while concepts like beach, man, sun, lake, mountain are biased towards the Waterbird class. This is the background bias that is known in the Waterbirds dataset, that ConBias successfully uncovers.

### 9.2    UrbanCars

In UrbanCars, the class labels (country car, urban car) are intentionally biased towards background and co-occurring objects. In Figure 11, we see that there exists an extreme imbalance betwee urban concepts such as driveway, traffic light towards urban cars, and country concepts such as forest road, field road, cow, horse towards country cars. These are exactly the background and co-occurring biases in the construction of the data, that ConBias successfully uncovers.

### 9.3    COCO-GB

The gender bias in COCO-GB has been extensively studied in [50]. In Figure 12, we show the extreme imbalance towards specific concepts in the MS-COCO dataset. Concepts such as baseball bat, sports ball, motorcycle, truck overwhelmingly correlate with images of men, which may be problematic for the classifier to learn.

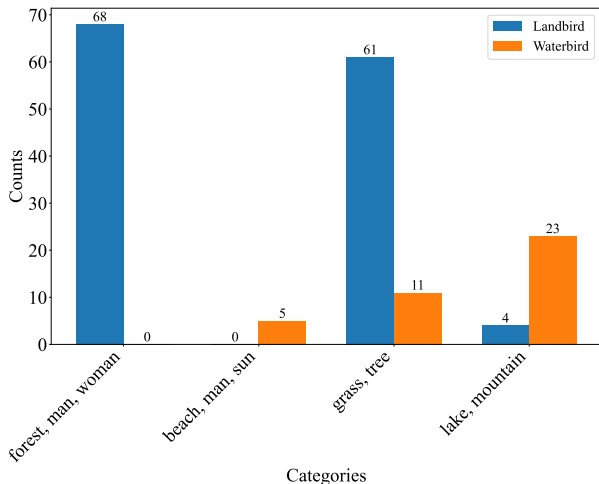

Figure 10: Extreme imbalance of particular concepts in Waterbirds dataset, as discovered by ConBias.

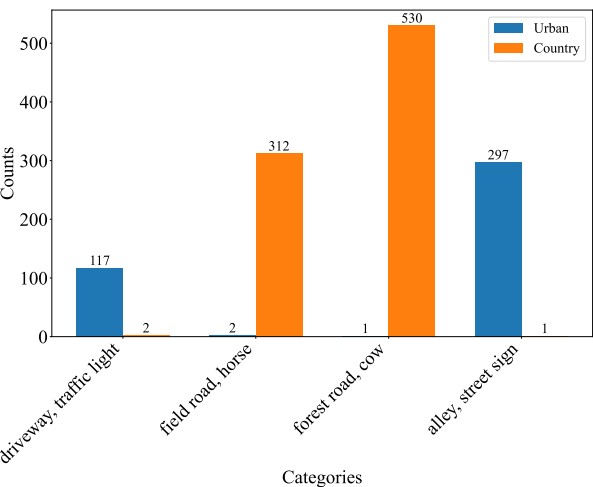

Figure 11: Extreme imbalance of particular concepts in UrbanCars dataset, as discovered by ConBias.

## 10 Generative Model

Here we present more details of our generative model. We use Stable Diffusion based inpainting, as illustrated in Figure 13. Given the prompt, we first generate an image using Stable Diffusion [44]. Next, using ground truth masks of the object, we paste the object at the foreground of the generated image. In this way, we preserve the original object in the image, which is a challenge for traditional image editing methods such as InstructPix2Pix. We believe the inpainting method is a more principled approach to synthetic image generation, particularly if the downstream task is classification in nature.

The generation process of a single image takes the followings as input: The sampled concept combination and the class for which this concept combination needs to be generated. The output of the generative model is the final image with the specified concepts in the background and an instance of the specified category in the foreground.

The process will first transform the concept list [concept 1, concept 2, ..., concept N] into a prompt: "a photo of concept 1, concept 2, ..., and concept N." The prompt is then passed into the text-to-image generation model (stable diffusion) to get the generated image as background. We apply a clip-score filtering after the generation process to only keep the images with a CLIP-score over 0.6 to make sure

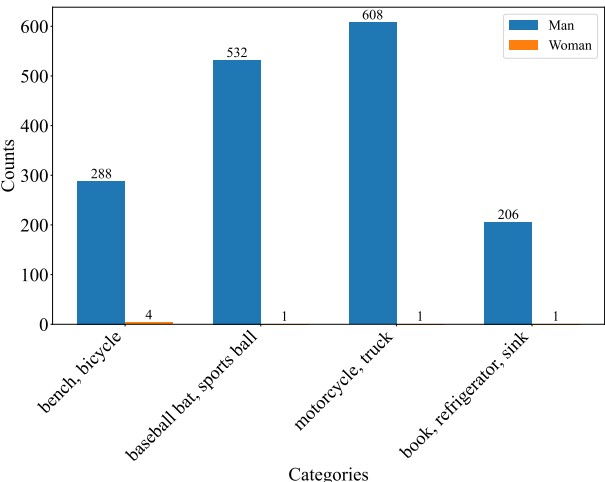

Figure 12: Extreme imbalance of particular concepts in MS-COCO dataset, as discovered by ConBias.

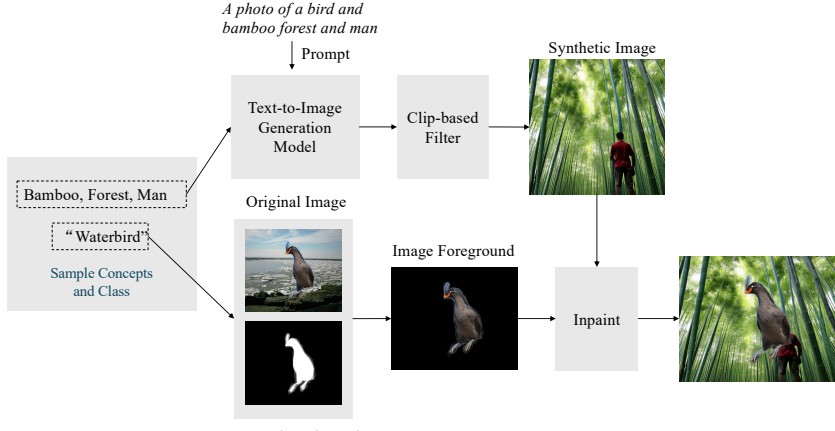

Figure 13: Image generation Pipeline: Given concepts to be upsampled as discovered ConBias, we sample the concept combinations and images from the class to be upsampled. We prompt Stable Diffusion for an image containing such concepts. We extract the object of interest using ground truth masks, and inpaint the object over the generated image. This ensures that the object features are not harmed during generation. We use a CLIP-based scoring filter to make sure the generated image contains the concepts requested in the prompt. We have found a score of 0.6 to be satisfactory as a threshold.

that the generated images can accurately represent the concept list. Next, the process will sample an image of the specified category from the original dataset, and use the mask to segment out the desired object. Finally, inpainting is performed to clip the desired object as foreground onto the generated image to obtain the final image.

## 11 Generated Images by ConBias

In this section we present examples of synthetic data generated by ConBias for Waterbirds, UrbanCars, and COCO-GB.

## 11.1 Waterbirds

In Figure 14 we present diverse images generated by ConBias for the two classes of Landbird and Waterbird. Due to the bias diagnosis stage where we found the overwhelming correlation between landbirds with land based backgrounds such as tree, forest, field, grass, etc, and waterbirds with water based backgrounds such as beach, ocean, boat, etc, ConBias was automatically able to decide which concept combinations to use to generate new, debiased images.

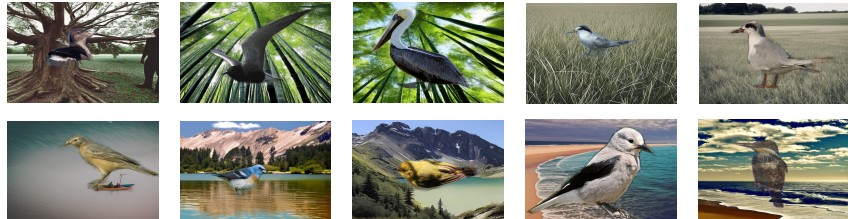

Figure 14: (Top) Generated images of waterbirds with land-based backgrounds. (Bottom) Generated images of landbirds with water-based backgrounds, as discovered by ConBias. Note the consistency in object preservation.

## 11.2 UrbanCars

In Figure 15 we present diverse images generated by ConBias for the two classes of Urban and Country cars. Due to the bias diagnosis stage, we were able to discover the overwhelming correlation between urban cars with urban based backgrounds such as gas station, driveway, alley, etc and urban co-occurring objects such as fireplug, stop sign, etc. Similarly, for country cars, we discovered bias towards country backgrounds such as desert road, field road, forest road, and, and country co-occurring objects such as cow, sheep, horse. As a result, ConBias helps generate urban cars with country based backgrounds and co-occurring objects, and vice versa.

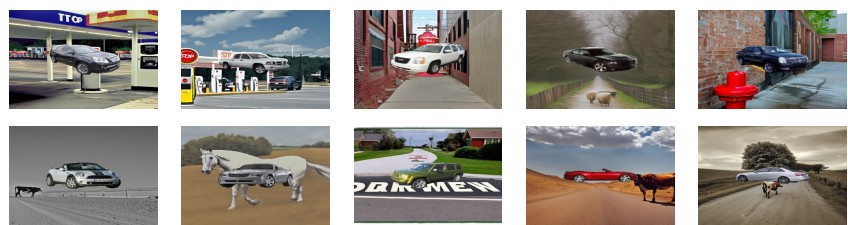

Figure 15: (Top) Generated images of country cars with urban-based backgrounds and co-ocurring objects. (Bottom) Generated images of urban cars with urban-based backgrounds and co-occurring objects, as discovered by ConBias. Note the consistency in object preservation.

## 11.3 COCO-GB

In Figure 16 we present diverse images generated by ConBias for the two classes of Man and Woman in COCO-GB. In this dataset, we were able to discover significant under-representation of women with respect to common, everyday objects in the MS-COCO dataset. Some examples include `skateboard, motorcycle, car, truck`, etc. These objects could have gendered assumptions and it is imperative for debiased datasets to have uniform representation across classes for such concepts.

We would also like to bring to the attention of our readers the successful nature of the inpainting procedure. We are able to consistently preserve the *class label* of interest in the synthetic images. This is imperative to ensure that the generative pipeline does not create unreasonable objects that make it infeasible for the classifier to learn.

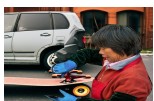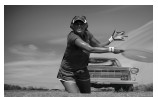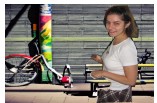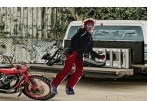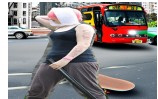

Figure 16: Generated images of COCO-GB using everyday, common objects that are discovered to be biased towards men by ConBias. Example concepts include `skateboard`, `motorcycle`, `truck`, `sports ball`, etc. Note the consistency in object preservation.

## 12 Confidence Intervals

In Table 7 we present the averaged results with standard deviations over three training runs. For both Waterbirds and UrbanCars, our improvements are large and significant. For COCO-GB, while originally did not observe statistically significant results, in the main paper we showed that increasing the number of images in $D_{aug}$ leads to significant improvements over the baselines.

Table 7: **State-of-the-art comparison on different datasets.** Results are averaged over three training runs. **CB**: class balanced split. **OOD**: out-of-distribution split. Binary class classification accuracy is used as the metric. CONBIAS outperforms previous approaches across multiple datasets. Standard deviations included.

| Method | Waterbirds [45] | | UrbanCars [22] | | COCO-GB [50] | |
|---|---|---|---|---|---|---|
| | CB | OOD | CB | OOD | CB | OOD |
| Baseline [17] | $67.1 \pm 0.5$ | $44.9 \pm 0.8$ | $73.5 \pm 0.6$ | $40.5 \pm 0.8$ | $58.5 \pm 0.7$ | $51.9 \pm 0.7$ |
| + RandAug [5] | $73.7 \pm 0.8$ | $60.2 \pm 0.7$ | $76.3 \pm 0.8$ | $46.1 \pm 0.9$ | $55.8 \pm 0.4$ | $50.2 \pm 0.6$ |
| + CutMix [60] | $67.9 \pm 0.7$ | $45.6 \pm 0.7$ | $74.4 \pm 0.7$ | $39.3 \pm 0.9$ | $57.4 \pm 0.5$ | $51.2 \pm 0.6$ |
| + ALIA [8] | $69.6 \pm 1.2$ | $48.2 \pm 1.0$ | $74.0 \pm 0.9$ | $42.5 \pm 0.9$ | $58.7 \pm 0.4$ | $\mathbf{52.4} \pm 0.6$ |
| + ConBias (ours) | $\mathbf{77.9} \pm 0.9$ | $\mathbf{69.3} \pm 0.8$ | $\mathbf{78.3} \pm 0.7$ | $\mathbf{52.9} \pm 0.7$ | $\mathbf{58.8} \pm 0.6$ | $51.4 \pm 0.4$ |

## 13 Compute Details

We trained all models on a single NVIDIA RTX A4000 and used PyTorch [35] for all experiments. With the early stopping cosine learning scheduled described in the main paper, we observed fast training times, with 90 minutes for three runs on Waterbirds and UrbanCars, and 180 minutes for three runs on COCO-GB.

## 14 Sampling Algorithm

The rebalance sampling algorithm 1 receives the concept graph as a concept co-occurrence matrix. The algorithm iterates through all cliques with an order of decreasing clique sizes to make sure we would not double compensate for the imbalance, e.g., 3-cliques would impact the already-balanced 2-cliques if we operate in a bottom-up fashion. For each iteration, it retrieves all cliques of concepts of size k along with their corresponding frequencies with each class. The algorithm identifies the maximum co-occurrence count among all classes for each combination and checks if any class is under-represented by comparing its count with the maximum. If a class is under-represented, the algorithm computes the number of synthetic samples needed to balance the representation and adds this information to the results list. This process continues for all combinations and classes until all clique sizes have been processed. The output of the algorithm is a list of queries specifying the class, concept combination, and the number of samples needed to balance the dataset.

## 15 Diagnosing ImageNet-1k

In this section we demonstrate the usefulness of ConBias in diagnosing spurious concepts in a more complex dataset, i.e. ImageNet-1k. Specifically, we investigate the classes *ambulance, beach wagon, sports car, limousine, minivan, jeep, convertible, cab*. These are associated with the superclass of Car. This approach can be used for any set of classes in the dataset. We use the open-source concept annotations available here.

**Algorithm 1** Rebalance Sampling

**Input:** $M_{occ}$ - Concept occurrence matrix as a dictionary
1: $results \leftarrow []$ ▷ Initialize result list for complement samples
2: $k \leftarrow$ the maximum size of concept cliques
3: **while** $k > 1$ **do**
4:      $combos \leftarrow M_{occ}[k]$ ▷ Get the concept combinations and counts of all $k$-cliques
5:      **for each** concept combination $C \in combos$ **do**
6:          $n_i \leftarrow$ number of co-occurrence between combo $C$ and class $i$
7:          $m \leftarrow \max(\{n_i\})$ ▷ Determine the maximum frequency among the classes
8:          **for each** class $i$ **do**
9:              **if** $n_i < m$ **then** ▷ If the class $i$ is under-represented w.r.t. combo $C$
10:                 $\hat{n}_i \leftarrow m - n_i$ ▷ Compute the number of samples to generate
11:                 $results \leftarrow results \cup (i, C, \hat{n}_i)$ ▷ Save generation query
12:              **end if**
13:          **end for**
14:      **end for**
15:      Update $M_{occ}[k']_{k'<k}$ with the generated samples
16:      $k \leftarrow k - 1$ ▷ Move to cliques with size smaller by 1
17: **end while**
**Output:** $results$ ▷ The full set of queries to generate

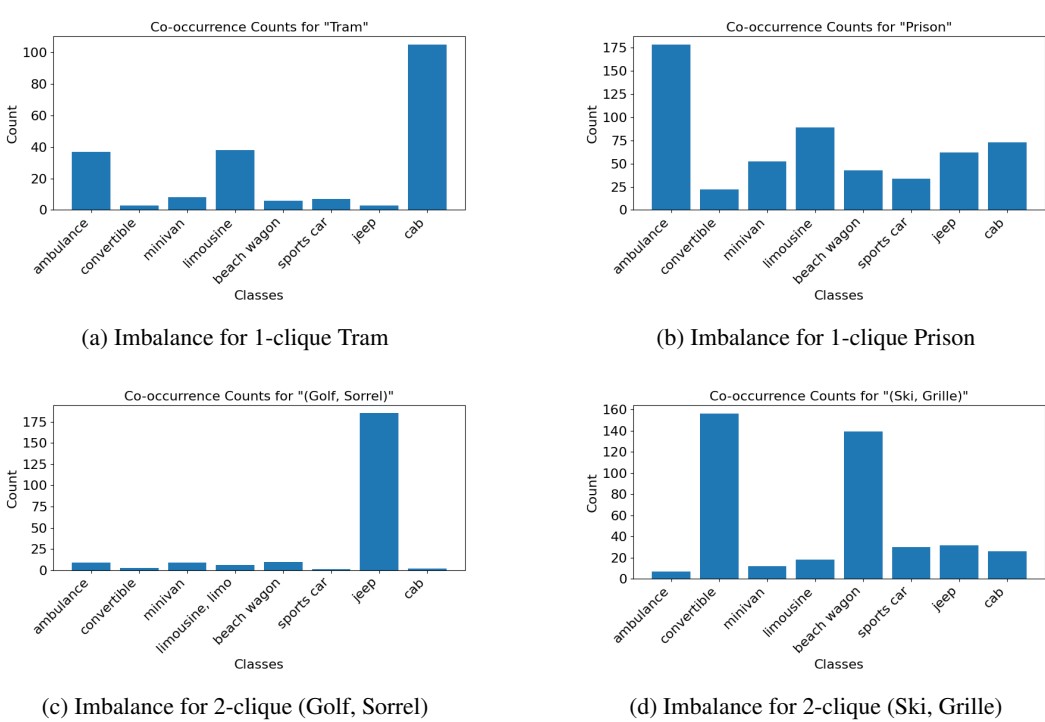

(a) Imbalance for 1-clique Tram

(b) Imbalance for 1-clique Prison

(c) Imbalance for 2-clique (Golf, Sorrel)

(d) Imbalance for 2-clique (Ski, Grille)

Figure 17: Concept imbalances for cars in ImageNet-1k

In Figure 17 we notice some interesting imbalances discovered by ConBias. For 1-clique concept combinations such as Tram and Prison, the dataset disproportionately contains images of cabs and ambulances respectively. For 2-clique concept combinations, the dataset disproportionately represents the jeep, convertible, and beach wagon classes. It is evident that such concepts are spurious when it comes to classifying a car type, but a strong imbalanced distribution would bias the classifier to pick up on spurious features. In this way, ConBias helps us uncover such biases, allowing for intervention in downstream tasks.

