# OpenReview forum: "Visual Data Diagnosis and Debiasing with Concept Graphs"
_NeurIPS.cc/2024/Conference — NeurIPS 2024 poster_

### Official Review · Reviewer_BpQx · 2024-07-10

**Soundness:** 3
**Presentation:** 3
**Contribution:** 3
**Rating:** 6
**Confidence:** 4

**Summary:**

The paper introduces a method for addressing the issue of inherent bias in data during the training process. This bias can lead to unreliable predictions from the model. The proposed method, called ConBias, is a new approach for identifying and mitigating Concept co-occurrence Biases in visual datasets. The method represents visual datasets as knowledge graphs of concepts, allowing for detailed analysis of misleading concept co-occurrences that reveal imbalances across the dataset. Additionally, the author demonstrates how a new clique-based concept balancing strategy can address these imbalances, resulting in improved performance on subsequent tasks. Extensive experiments show that applying data augmentation based on the proposed method leads to significant improvements in generalization performance across various datasets when compared to state-of-the-art methods.

**Strengths:**

The paper is well-written and explains the method thoroughly. The authors' experiments on various common datasets demonstrate the method's effectiveness and the comparisons to a prior art show that it achieves state-of-the-art performance. The paper includes a thorough analysis of the method's components, justifying its contributions.

**Weaknesses:**

In the paper, I missed the ablation study of the authors' knowledge graphs design, the impact of using a different number of graph layers, knowledge graph architectures, different initializations, etc.

**Questions:**

https://arxiv.org/abs/2310.04562 proposes "ULTRA" KD foundation graph pre-training. Can using this method benefit the proposed graph structure initialization?

**Limitations:**

The authors have properly addressed the proposed method limitations.

---

> ### Author Rebuttal · Authors · 2024-08-06
>
> We thank the reviewer for the positive comments.
>
> **Clarification on the ablation studies:**
> Since the concept graph phase in ConBias is not a learning mechanism, we circumvent the need for initialization or training strategies. This is beneficial as we achieve both controllability and interpretability of the bias diagnosis process. The imbalanced cliques provide an intuitive mechanism to diagnose the spurious correlations in the data. With regard to ablation studies, since the augmented data generation phase is a learning mechanism (we use stable diffusion with a CLIP based filter), here we present ablation results (Table 3). We show that using a different generative model such as IP2P still results in significant improvements over the baselines. This result demonstrates that it is actually the bias diagnosis and clique balancing strategy of ConBias that proves to be the true novelty.
>
>
> **Graph structure initialization:**
> We thank the reviewer for referring to the Ultra-KD paper. Broadly, such an approach excites us immensely, where a knowledge graph encapsulates deep information about the data. For instance, in lines 317 and lines 324, we mention the potential of future work in developing more novel graph structures that can capture diverse biases beyond object co-occurrences. Foundational knowledge graphs with rich relational information may prove to be the next logical step in dataset diagnosis and debiasing. We will add this discussion to the final version, and thank the reviewer for raising this idea.

---

> > ### Comment · Reviewer_BpQx · 2024-08-11
> >
> > Dear Authors,
> >
> > I have reviewed your responses to my and other reviewers' concerns, and I am satisfied with them. Therefore, I am keeping my original score.

---

### Official Review · Reviewer_TVhQ · 2024-07-13

**Soundness:** 3
**Presentation:** 4
**Contribution:** 2
**Rating:** 5
**Confidence:** 3

**Summary:**

The authors present ConBias, which is a novel approach for diagnosing and de-biasing co-occurrence bias in datasets. Unlike previous works, ConBias addresses both diagnosis and de-biasing / balancing strategy, to target the data augmentation specifically to address spurious concept co-occurrences. The main components of ConBias are as follows:
1. New concept graph-based analysis framework which involves occurrences of common concepts and classes in the data in a graph-form to identify spurious correlations
2. Identify main cliques that occur across all classes, and determine the frequency of occurrence with respect to each classes to find spuriously correlated concepts
3. re-balance the dataset based on the imbalanced concept cliques. specifically, generate new backgrounds for the objects using an inpainting-based augmentation.

Then they perform a thorough evaluation of their new re-balanced dataset across various evaluation protocols including the class-balanced test data and the OOD test data, as well as other metrics for shortcut learning. The results show that ConBias consistently outperforms other existing baselines. The authors also show ablations to justify their design decisions around the graph structure, as well as the generative model choice.

**Strengths:**

Originality: The work demonstrates originality by introducing targeted dataset-debiasing based on concept co-occurrence diagnosis within the dataset augmentation space. Unlike previous approaches that often debias without diagnosing or only diagnose biases within datasets without suggesting effective debiasing methods directly to the data, this method focuses on improving the robustness of binary classification tasks through meticulous dataset examination.

Quality: The paper presents a novel approach that consistently outperforms existing debiasing augmentation methods. The experimental setup spans multiple datasets across various domains, providing robust justification for the proposed approach. Results are clearly presented, demonstrating reasonable improvements in performance metrics.

Clarity: The paper is well-written and logically organized, making it easy to follow the ConBias pipeline. Graphs and illustrations are informative, enhancing understanding of each step and experimental setup.

Significance: This work introduces a debiasing method aimed at reducing overfitting to spurious correlations. This is significant as it can improve the reliability of vision tasks across various applications, especially in domains where robust performance on out-of-distribution (OOD) cases is crucial, such as healthcare diagnosis

**Weaknesses:**

The binary classification test bench used in the experiments may be insufficient in scope to fully demonstrate the potential of the proposed method. While the results show promise, it remains unclear if the method's performance scales effectively to multiple sets of classes, which is crucial for assessing its broader impact and significance. Moreover, in scenarios with multiple sets of classes, there exist spurious correlations across different subsets that require a more sophisticated and targeted sampling strategy than what the current approach, which extracts shared cliques across all classes, provides.

In addition, I believe further justification or benchmarking against debiasing methods in the feature space is necessary. Although this work pioneers combined diagnosis and debiasing in the dataset space, previous research has focused on diagnosing and debiasing models at training time or in feature spaces. Earlier methods required dataset bias labels, while recent advancements explore shortcut learning without explicit labels. The paper briefly mentions choosing the dataset space for enhanced interpretability but does not fully justify the computational trade-offs involved in generating balanced data as well as concept labels. The work would benefit significantly from additional benchmarking against state-of-the-art feature space analysis and debiasing methods where they are able to demonstrate improvements or providing a detailed rationale for its focus on the dataset debiasing.

**Questions:**

Related to the preceding section, I have several inquiries for the authors:

1. Could you provide insights into how this method might scale effectively in a multi-class classification scenario? What are your expectations regarding its performance scalability?
2. Could you clarify why additional evaluation metrics were exclusively presented for UrbanCars and not for Waterbirds or Coco-GB datasets?
3. How precisely is the upsampling conducted concerning different sets of cliques? While I understand the additional image count aligns with ALIA's methodology for fair comparison, it remains unclear how many images this comprises, how various co-occurring object sets contribute to the augmentation dataset distribution, and the criteria used in making these decisions.

**Limitations:**

Yes, the authors sufficiently address limitations and societal impact in the final section of their paper.

---

> ### Author Rebuttal · Authors · 2024-08-06
>
> Thank you for the encouraging review and the detailed feedback.
>
> **Extension to multi-class tasks and scalability:**
> ConBias can be conveniently extended to the multi-class setup. For this, we note the definitions in line 142 and line 146. While the common clique computation remains the same, for the imbalanced clique computation, instead of taking the pairwise absolute difference between classes, we can use a different metric such as variance and entropy to estimate the degree of imbalance among multiple classes. Functionally, the imbalanced set would contain the same information, i.e. how disproportionately is one concept combination (1-clique, 2-clique, etc.) represented with respect to a particular class relative to the other classes? In addition, in the attached PDF for the rebuttal (Figure 1), we also show how ConBias can diagnose biases in a complex multi-class dataset such as Imagenet-1k.
> We will add this discussion to the final version and thank the reviewer for the insightful question.
>
> For scalability,  we invite the reviewer to refer to Section H in the Supplementary section for details on runtime. In general, given $K$ classes and $C$ concepts, the graph clique enumeration is expected to grow in $O(exp(|K+C|)$. However, in practice, we find that constraining clique sizes to $k<=4$ leads to interpretable bias combinations, with no significant effect of the exponential runtime. We agree that this is a heuristic, and more efficient clique enumeration methods can be developed (lines 314-315). We will update the paper to reflect this point raised by the reviewer.
>
> **Comparison to debiasing methods based on the feature space:**
> ConBias is a data-centric method rather than a model-centric method. We wish to intervene in the data directly rather than using a specific model as a proxy (lines 80-83). Once the biases are diagnosed, we can controllably and reliably generate fair data that improves model debiasing and generalization capabilities on downstream tasks.
>
> A direct benchmarking against feature based methods would not be feasible since it is difficult to control for the effect of the augmented dataset. This is why our baselines, such as ALIA, also do not compare against feature-based debiasing methods. To ensure a fair comparison for data debiasing techniques, we need to ensure that the effect of adding data to the training set is adjusted for. In a pure feature debiasing based setup, this would be hard to control. However, the reviewer raises an important point about the dependence between data and feature debiasing. We agree that these two are not mutually exclusive. In fact, this is why we present results in Figure 6 where we visualize the classifier feature heatmaps after retraining. We demonstrate that the features learned by the classifier are indeed the core features, and not spurious features. On each dataset in Figure 6, we show that the classifier focuses on the object of interest (the core features), and not the background/gender/co-occurring object - which are precisely the spurious cues.
>
>
> **Evaluation metrics:**
> In Table 4, we evaluate the multi-shortcut abilities of ConBias. Multiple shortcut mitigation is a recent robustness problem introduced by Li, Zhiheng, et al. "A whac-a-mole dilemma: Shortcuts come in multiples where mitigating one amplifies others." Proceedings of the IEEE/CVF Conference on Computer Vision and Pattern Recognition. 2023. UrbanCars is the standard dataset for evaluating multiple shortcut robustness, with BG-Gap, CoObj-Gap, and BG+CoObj-Gap as the evaluation metrics. Both Waterbirds and COCO-GB are single shortcut datasets, and as such, these metrics do not apply to these datasets.
>
> **Upsampling:**
> The rebalance sampling algorithm receives the concept graph as a concept co-occurrence matrix. The algorithm iterates through all cliques with an order of decreasing clique sizes to make sure we would not double compensate for the imbalance, e.g., 3-cliques would impact the already-balanced 2-cliques if we operate in a bottom-up fashion. For each iteration, it retrieves all cliques of concepts of size k along with their corresponding frequencies with each class. The algorithm identifies the maximum co-occurrence count among all classes for each combination and checks if any class is under-represented by comparing its count with the maximum. If a class is under-represented, the algorithm computes the number of synthetic samples needed to balance the representation and adds this information to the results list. This process continues for all combinations and classes until all clique sizes have been processed. The output of the algorithm is a list of queries specifying the class, concept combination, and the number of samples needed to balance the dataset. We will include a pseudo-code in the final version for better readability.

---

> > ### Comment · Area_Chair_KbYF · 2024-08-13
> >
> > Reviewer TVhQ, do you have any additional questions or feedback?

---

> > > ### Comment · Reviewer_TVhQ · 2024-08-13
> > >
> > > Thank you authors for the detailed rebuttal and clarification of some of my concerns. After reviewing the rebuttal as well as the other reviews, I will keep my original score.

---

### Official Review · Reviewer_XqYX · 2024-07-13

**Soundness:** 2
**Presentation:** 3
**Contribution:** 2
**Rating:** 4
**Confidence:** 4

**Summary:**

This paper points out inherent biases in visual dataset. To address this issue, the paper proposes a new framework called ConBias, which proceeds in three steps: (1) Concept Graph Construction, (2) Concept Diagnosis, and (3) Concept Debiasing. Using concept metadata in the dataset, concept graph is constructed. After identifying imbalanced clique sets, ConBias utilizes Stable Diffusion to generate images containing under-represented concepts across classes. The experimental results show the effectiveness of the proposed framework to mitigate dataset biases.

**Strengths:**

1. The paper proposes a new concept graph-based framework, which is easy to construct using metadata and diagnose biases in visual datasets.
2. A new approach to generate unbiased images using stable diffusion robustly enhances the overall performance.
3. The experimental results across different datasets show reasonable performances compared to the state-of-the-art method.

**Weaknesses:**

1. The main concern is that the proposed framework should require metadata to construct a graph. It can be a problem to apply this method directly in many cases where visual datasets do not include any metadata.
2. I do not agree that the existing approach utilizing LLMs is no reliable way, since (1) there is also a possibility that the given metadata has wrong or biased information, and (2) as recent LLMs become powerful, the quality of domain descriptions is reliable (rather automatically improving the quality as time goes by). To sum up, there is no guarantee that the proposed method is always less biased compared to the existing method.
3. The processes to construct graph and diagnose bias do not look technically novel. The overall process is primarily based on a counting mechanism, not a learning-based method.
4. The authors tested the proposed framework on three benchmark datasets. However, the used datasets look like toy experiments, i.e., all the tasks are binary classification tasks and easy to identify biases. I wonder that the proposed method still constructs and detects inherent, complex biases in large and multi-class tasks, e.g., ImageNet.
5. The experimental results in the tables are quite different from the values displayed in the ALIA paper [8]. If it is true, can you explain the reasons?
6. There are no comparisons about computational complexity between the existing methods and the proposed one. Please clarify how the costs are changed per the number of total concepts.

**Questions:**

1. Please describe total numbers of generated graph nodes and edges.

**Limitations:**

The authors address limitations and broader impact well in the paper.

---

> ### Author Rebuttal · Authors · 2024-08-06
>
> We thank the reviewer for the valuable comments.
>
> **Metadata:**
> As we discuss in the main paper (lines 324-325), we assume the availability of reliable ground truth concept sets. Such annotations already exist for the datasets we investigate - Waterbirds, UrbanCars, and COCO-GB. We agree that unreliable ground truth concept sets would hinder generalization abilities, but this assumption is not dissimilar to the assumption of reliable ground truth labels in classification tasks. Moreover, the reliance on ground truth concept sets, sometimes referred to as concept banks, have also been considered in [A]. Ground truth concept sets serve as auxiliary knowledge bases and provide human level interpretability to the task at hand. We look forward to using partially available concept sets in future work (lines 324-325). We will update the paper to reflect this important point raised by the reviewer.
>
> [A] Wu, Shirley, et al. "Discover and cure: Concept-aware mitigation of spurious correlation." In ICML, 2023.
>
>
> **LLMs:**
> With regard to the reviewer’s first point, we do not argue in the paper that LLMs are not reliable for metadata, and we agree with the reviewer that LLMs are useful for a variety of tasks. The metadata that we use comes from ground truth concepts that are available in all the datasets tested in our work. We do not use LLMs to extract metadata for graph construction.
> Instead, we argue that LLMs are unreliable in the generation of fair, balanced data (lines 36-39). This is an issue that has been acknowledged by the authors of ALIA as well (please refer to Section 6 in the ALIA paper). This is the issue we overcome with ConBias: Instead of using an off-the-shelf LLM to generate diverse image data, we use a controllable and interpretive knowledge graph that encodes the class-concept imbalances in the data.
>
> **Novelty:**
> The core intuition of ConBias is that we want a controllable and interpretable way to generate debiased data. In a learning based framework, these two requirements are compromised since a gradient based optimization technique relies on parameters that may deem the model to be a black box.
>
> ConBias is not simply a statistical co-occurrence counting mechanism. In fact, in lines [87-90], we mention that there are diagnostic tools that exist today that compute such object co-occurrence statistics. The novelty of our method is in the creation of a graph that encapsulates multiple class-concept co-occurrences. The analysis of the class-concept graph cliques is crucial to the graph construction. In Table 2, we show the benefit of using the graph structure as opposed to a simple counting mechanism based on statistical co-occurrences. Additionally, our method is novel as to the best of our knowledge, no existing works leverage a concept co-occurrence based graph to simultaneously diagnose and debias datasets. Our results, demonstrated in Table 1-4, confirm the usefulness of the graph clique balancing strategy.
>
> Broadly, we believe that if we wish to debias datasets, we need human-level intuition on what concepts are biased, and what concepts need to be debiased in a principled fashion, corresponding to human-level intuition. ConBias provides this intuition (please refer to Figure 3 and Figure 4 in the main paper, and Section D in the supplementary section). A black-box learning based approach, on the other hand, would hinder these capabilities.
>
>
> **Extension to multi-class tasks:**
>
> Since the reviewer raised the example of Imagenet, in the attached PDF above, we present new results on diagnosing multiple classes in Imagenet-1k. We discovered some interesting spurious correlations uncovered by ConBias. As such, ConBias can be seamlessly used for datasets of increasing complexity. We will update the paper to include these results.
>
> Waterbirds and UrbanCars, while binary, are the de facto datasets used in the literature for evaluating single-shortcut and multi-shortcut learning in classifiers. These are the most commonly used datasets for benchmarking purposes. COCO-GB is a subset of the MS-COCO dataset that includes real-world data consisting of humans and everyday objects in the wild.
>
> We include the motivations behind the datasets and other details in Section 4.1 and Supplementary Section B. For the reviewer’s convenience, we also include three references here that may clarify the use of these datasets:
>
> - Li, Zhiheng, et al. "A whac-a-mole dilemma: Shortcuts come in multiples where mitigating one amplifies others." In CVPR. 2023.
>
> - Sagawa, Shiori, et al. "Distributionally Robust Neural Networks." In ICLR 2020.
>
> - Tang, Ruixiang, et al. "Mitigating gender bias in captioning systems." In WWW. 2021.
>
> **Results of ALIA:**
> We invite the reviewer to refer to Section G of the Supplementary material, where we present the confidence intervals in addition to the averaged results in Table 1. The results presented are within the confidence intervals as presented in the ALIA paper, who also average results over three runs.
>
>
> **Computational complexity:**
> We invite the reviewer to refer to Section H in the Supplementary section for details on runtime. In general, given $K$ classes and $C$ concepts, the graph clique enumeration is expected to grow in $O(exp(|K+C|)$. However, in practice, we find that constraining clique sizes to $k<=4$ leads to interpretable bias combinations, with no significant effect of the exponential runtime. We agree that this is a heuristic, and more efficient clique enumeration methods can be developed (lines 314-315). We will update the paper to reflect this point raised by the reviewer.
>
>
> **Total number of graph nodes and edges:**
> For Waterbirds, there are 66 nodes and 865 edges in the graph. For UrbanCars, the graph contains 19 nodes and 106 edges. For COCO-GB, there are 81 nodes and 2326 edges in the graph. We will update the paper to include these additional details.

---

> > ### Comment · Reviewer_XqYX · 2024-08-13
> >
> > Thank you for the responses from the authors. Some ambiguities I raised have been addressed. However, I still have concerns about the authors’ responses, so I keep my initial rating. For example, regarding metadata, I mean the graph construction in the inference time, where no label exists. I know that the datasets the authors used in the manuscript have metadata, but in many real-world scenarios, most data samples are unlabeled, where ConBias cannot apply. Moreover, I think the authors misunderstood my point in LLMs. I did not mean that LLMs are not reliable for metadata, or the authors use LLMs to extract metadata for graph construction.

---

> ### Author Response · Authors · 2024-08-13
>
> We thank the reviewer for the comments. Here, we address the points on metadata and LLMs.
>
> **Metadata**
>
> In the absence of ground truth metadata, one can leverage large multimodal models (LMMs), for instance in segmentation or open-vocabulary detection, and generate such concepts. However, there remains the possibility of noisy metadata that may contain unreliable artefacts, in addition to the biases within these models themselves. In this work, we restrict ourselves to available, high-quality, ground truth concepts, to showcase the usefulness of our framework. Pursuing open-vocabulary models to generate metadata is an interesting direction of future work. In general, ConBias is not constrained by how the metadata is obtained, but in the quality of metadata obtained. Future developments in LLMs/LMMs that can generate high quality concept metadata can be seamlessly integrated into the ConBias framework. We will be sure to update the paper with this discussion.
>
> Our core contribution with ConBias is not the metadata stage, which we assume to be available and high-quality, similar to other works in the past [A, B]. Our core contribution is the diagnosis and debiasing of datasets with ConBias, which leads to significant improvements on multiple datasets with respect to the current state-of-the-art. As requested by the reviewer, we have also provided diagnosis results on a more complex dataset such as Imagenet-1k.
>
> [A] Wu, Shirley, et al. "Discover and cure: Concept-aware mitigation of spurious correlation." In ICML, 2023.
>
> [B] Lisa Dunlap, Alyssa Umino, Han Zhang, Jiezhi Yang, Joseph E Gonzalez, and Trevor Darrell. Diversify your vision datasets with automatic diffusion-based augmentation. Advances in Neural Information Processing Systems, 36, 2024
>
> **LLMs**
>
> We reiterate that, as mentioned in lines 36-39, relying on LLMs to generate diverse, unbiased descriptions is problematic since LLMs themselves may be biased, and such generation is not controllable. This issue of relying on LLMs has also been addressed in the ALIA paper (Section 6) [A], and it is precisely this issue that we fix with ConBias. By leveraging the concept graph, ConBias can generate a debiased dataset in a controlled and interpretable manner.
>
> [A] Lisa Dunlap, Alyssa Umino, Han Zhang, Jiezhi Yang, Joseph E Gonzalez, and Trevor Darrell. Diversify your vision datasets with automatic diffusion-based augmentation. Advances in Neural Information Processing Systems, 36, 2024

---

> > ### Comment · Area_Chair_KbYF · 2024-08-13
> >
> > Reviewer XqYX, do you have any additional questions or feedback?

---

### Official Review · Reviewer_xMok · 2024-07-15

**Soundness:** 2
**Presentation:** 3
**Contribution:** 2
**Rating:** 5
**Confidence:** 3

**Summary:**

This paper introduces a concept graph-based framework to diagnose and mitigate biases in visual datasets by representing datasets as knowledge graphs of object co-occurrences. The approach involves constructing a concept graph, diagnosing concept imbalances, and debiasing by generating images with under-represented concept combinations. This method enhances dataset balance and improves generalization performance across multiple datasets.

**Strengths:**

	The paper presents a structured and controllable method for diagnosing and mitigating spurious correlations by representing datasets as knowledge graphs of object co-occurrences.
	Experimental results demonstrate that balanced concept generation in data augmentation enhances classifier generalization across multiple datasets, outperforming baseline methods.
	This manuscript is written well, it’s easy to read and follow.

**Weaknesses:**

This manuscript seems to be an engineering technical report about the data augment and the technical contribution is weak for an academic paper. The method of constructing a knowledge graph to determine object co-occurrences is overly direct; statistical analysis of objects and frequency comparison can achieve similar results.
•  What is the advanced intuition behind constructing a knowledge graph to determine object co-occurrences? How does it compare to previous methods in terms of advantages?
•  The paper lacks details on the concept set annotations. How might the quality of these annotations affect the performance of the proposed method?
•  Why are the results of ALIA on Table 1 difference with ALIA[1].
[1] Lisa Dunlap, Alyssa Umino, Han Zhang, Jiezhi Yang, Joseph E Gonzalez, and Trevor Darrell. Diversify your vision datasets with automatic diffusion-based augmentation. Advances in Neural Information Processing Systems, 36, 2024

There is a typographical error on line 198: a space is missing after the period.

**Questions:**

see above

**Limitations:**

Yes, the authors discuss about limitations in section 5.

---

> ### Author Rebuttal · Authors · 2024-08-06
>
> We thank the reviewer for the positive feedback.
>
> **Intuition and advantage of constructing knowledge graph:**
>  ConBias is not simply a statistical co-occurrence counting mechanism. In fact, in lines [87-90], we mention that there are diagnostic tools that exist today which compute such object co-occurrence statistics. The novelty of our method is in the creation of a graph that encapsulates multiple class-concept co-occurrences. The analysis of the class-concept graph cliques is crucial to the graph construction. In Table 2, we show the benefit of using the graph structure as opposed to a simple counting mechanism based on statistical co-occurrences. Additionally, our method is novel as to the best of our knowledge, no existing works leverage a concept co-occurrence based graph to simultaneously diagnose and debias datasets. The advantages over previous methods is in the controllability and interpretability of debiased data generation. Our results, as demonstrated in Table 1 and Table 4, show significant advantages in generalization over previous approaches.
>
>
> **Details of concept set annotations:**
> We described the details for all the datasets used in Section B and Section C of the appendix and the supplementary materials. In Table A2 in appendix, we detailed the list of concept sets: the Waterbirds dataset has 64 unique concepts; the UrbanCars dataset has 17 unique concepts; the COCO-GB dataset has 81 unique concepts. All the concepts are from the MS-COCO dataset. We will incorporate these details into the main paper in the final version.
>
>
> **Effect of the quality of the annotations:**
> We assume the concept sets to be available for graph construction. One can interpret this availability as the same as the assumption that ground truth labels are useful for classification tasks. Just as unreliable ground truth labels can hinder classifier performance, the concept sets also need to be of high quality and reliability to ensure strong generalization. We mention this assumption in the final section of the main paper, and are looking forward to future work that investigates unreliable, partially available concept sets. We will update the paper to clarify this important point raised by the reviewer.
>
>
> **Results of ALIA:**
> We invite the reviewer to refer to Section G of the appendix, where we present the confidence intervals in addition to the averaged results in Table 1. The results presented are within the confidence intervals as presented in the ALIA paper, who also average results over three runs.
>
> **Typos in L198:**
> Thank you and we will correct it in the final version.

---

> > ### Comment · Reviewer_xMok · 2024-08-13
> >
> > Thanks for the rebuttal. It helps me to solve my concerns. I keep my score: bdl accept for this work.

---

### Author Rebuttal · Authors · 2024-08-06

We thank the reviewers for providing valuable feedback on our work. We are glad that they found our idea "original" (R3); our diagnosing and debiasing method "structured and controllable" (R1), "novel, significant" (R3); our graph-based framework "a new approach to generate unbiased images" (R2), "robust" (R2); our graphs and illustration "informative" (R3); our experiments and justification "robust" (R3), "thorough" (R4); our paper "well-written" (R1, R4).

In this rebuttal, we separately address each reviewer’s remaining comments as follows. Also, the attached PDF contains results from diagnosis experiments on Imagenet-1K, as requested by reviewer R2. We will incorporate all the feedback to the best of our abilities.

R1: Reviewer **xMok**; R2: Reviewer **XqYX**; R3: Reviewer **TVhQ**; R4: Reviewer **BpQx**.

---

### Decision · Program_Chairs · 2024-09-25

**Decision:**

Accept (poster)

**Comment:**

The reviews are mixed, but most reviewers are leaning to accept. The primary concern that remains across several reviewers is that the proposed approach relies heavily on the existence of accurate and high-quality metadata, and was only tested on less-complex datasets without realistic bias. The ACs agree that these are limitations to the general application of this approach, but there are many domains (for example many areas of science) where curated metadata is available for all datapoints. Thus, while the method proposed may not be useful for all data, the contribution would be potentially impactful for some subareas and the ACs have decided to recommend accept.